# Scalable Fabrication of Metallopolymeric Superstructures for Highly Efficient Removal of Methylene Blue

**DOI:** 10.3390/nano9071001

**Published:** 2019-07-11

**Authors:** Meirong Zhou, Tianyu Yang, Weibin Hu, Xiaohong He, Junni Xie, Pan Wang, Kun Jia, Xiaobo Liu

**Affiliations:** Research Branch of Advanced Functional Materials, School of Materials and Energy, University of Electronic Science and Technology of China, Chengdu 611731, China

**Keywords:** metallopolymeric superstructures, self-assembly, methylene blue, adsorption

## Abstract

Metallopolymeric superstructures (MPS) are hybrid functional materials that find wide applications in environmental, energy, catalytic and biomedical-related scenarios, while their fabrication usually suffers from the complicated polymerization between monomeric ligands and metal ions. In this work, we have developed a facile one-step protocol to fabricate metallopolymeric superstructures with different morphology including nanospheres, nanocubes, nanorods, and nanostars for environmental remediation application. Specifically, we have firstly synthesized the amphiphilic block copolymers (BCP) bearing hydrophobic aromatic backbone and hydrophilic pendent carboxylic/sulfonic groups, which have been subsequently transformed into MPS via the metal ions mediated self-assembly in mixed solution of dimethylformamide (DMF) and H_2_O. Based on SEM, FTIR, XRD and XPS characterization, we have revealed that the fine morphology and condensed structures of MPS can be modulated via the metal ions and BCP concentration, and the obtained MPS can be employed as efficient adsorbents for the removal of methylene blue with maximum adsorption capacity approaching 936.13 mg/g.

## 1. Introduction

Metallopolymeric superstructures, such as metal organic framework (MOF), metal coordination polymer (MCP), porous coordination polymer (PCP), etc., have witnessed increasing research interests in recent years mainly due to their versatile functionality and easy modification [1,2]. Generally, the metallopolymeric superstructures (MPS) have been prepared via the complicated polymerization between specifically designed monomeric ligands and metal ions, resulting in a great challenge of scalable fabrication. Alternatively, it would be very interesting to fabricate MPS directly using pre-synthesized polymers and metal ions as building blocks [3], which would effectively expand the practical application of the MPS, especially for adsorption removal of organic dyes in environmental remediation, which normally requires large quantity of adsorbent materials.

Water pollution is one of the main environmental challenges worldwide, which has attracted increasing research attention, and the pollution from organic dyes intensively used in textile industries is a major concern and poses serious threats to ecosystems [4]. For instance, it has been reported that the cationic dye of methylene blue (MB) may be responsible for permanent injury to the eyes of animals and human, cause short periods of difficult breathing upon inhalation, and nausea, vomiting, excessive sweating, mental confusion, urination pain as well as methemoglobinemia through oral ingestion [5,6]. Given these harmful environmental influences, MB should be efficiently removed prior to entering into water ecosystem [7]. Various biological, chemical, and physical technologies, including adsorption [8], biosorption [9], advanced oxidation [10], membrane filtration [11], flocculation/coagulation [12], and ozonation [13] have been widely explored to purify the dyes wastewater [14,15,16,17]. Each of the above methods has their own pros and cons, and the main concern is how to remove organic dyes economically for the environmental research and application community [18,19].

The adsorption method, utilizing physical and chemical interactions such as van der Waals forces, electrostatic attraction, hydrogen bonding, chelation, conjugation, and coordination to achieve aggregation of the adsorbates on the surface of the adsorbent [20,21], is an environmentally friendly technique that can be operated at low cost [22,23,24,25,26]. Many studies have been widely conducted to investigate the use of low-cost adsorbents such as wood shavings silica [27], fly ash [28], steel-plant slag [29], peat [30], china clay [31], and bentonite [32] for dyes removal. Unfortunately, these low-cost adsorbents typically have low adsorption capacity, which limits their widespread using. In recent years, a variety of advanced metallopolymeric superstructures (MPS) have been developed as efficient adsorbents for dyes removal due to their large specific surface area, excellent mechanical strength, adjustable surface chemistry, tunability of pore structure and feasible regeneration under mild conditions, which will effectively solve the limitations of conventional adsorbents [33,34,35,36]. However, the fabrication of these MPS is time-consuming and difficult to prepare in large scale, which would hinder their practical applications in environmental remediation.

Polyarylene ether nitrile (PEN) is high performance engineering thermoplastic that possess promising thermostability, mechanical properties, and versatile processability, mainly due to their unique macromolecular structures containing aromatic backbone and polar pendent cyano groups. The rigid backbone structure renders the intrinsic aggregation tendency of PEN macromolecular chains, which can be further enhanced by the strong intramolecular interaction derived from presence of pendent polar groups. Recently, we have synthesized a range of amphiphilic block copolymer by manipulation of backbone structure as well as pendent groups of conventional PEN, leading to fabrication of different polymeric superstructures via microemulsion self-assembly for organic dyes removal, metal ions detection, as well as multimodal in vitro bio-imaging [37,38]. However, the fabrication of PEN superstructures in these previously published works is the exquisite self-assembly in immiscible microemulsion system, which is still difficult to scale up for industrial application. Therefore, we herein have synthesized a novel amphiphilic PEN block copolymer (amPEN) containing higher density of pendent carboxyl/sulfonic groups and are able to fabricate MPS with different morphology via a facile solvent-exchange-induced self-assembly in the presence of different metal ions. More importantly, we found that the obtained MPS exhibit quite competitive adsorption capacity for MB removal.

## 2. Experimental

### 2.1. Chemicals and Materials

Analytical reagents (AR) of hydroquinone monosulfonic acid potassium salt (SHQ), phenolphthalin (PPL), 2, 6-difluorobenzonitrile (DFBN), *N*,*N*-dimethylformamide (DMF), *N*-methyl pyrrolidone (NMP), toluene, and potassium carbonate (K_2_CO_3_) were purchased from Chengdu Haihong Chemical (Chengdu, China). Analytical reagents (AR) of lead nitrate (Pb(NO_3_)_2_), calcium chloride (CaCl_2_), cobaltous chloride hexahydrate (CoCl_2_·6(H_2_O)), ferrous chloride tetrahydrate (FeCl_2_·4(H_2_O)), Ferric chloride hexahydrate (FeCl_3_·6(H_2_O)), chromium chloride hexahydrate (CrCl_3_·6(H_2_O)), and zirconium chloride (ZrCl_4_) were provided by Adamas-beta (Shanghai, China). Methylene blue (MB) was acquired from Guangfu Fine Chemical Institute (Tianjin, China). All chemical reagents were used without further treatment and deionized water was used throughout the experiments. 

### 2.2. Synthesis of Amphiphilic Polyarylene Ether Nitrile Block Copolymer (amPEN)

The synthesis of amPEN was performed according to our previous studies [39]. The weight average molecular weight (Mw) of the amPEN was measured on a gel permeation chromatography (GPC, Tosoh, HLC-8320, Tokyo, Japan) to be 129,547 g/mol.

### 2.3. Preparation of Metallopolymeric Nanostructures (MPS)

Firstly, 50 mg synthesized amPEN was dissolved into 1 mL DMF to form a homogenous solution (50 mg/mL). Simultaneously, several metal salt including Pb^2+^, Ca^2+^, Co^2+^, Fe^2+^, Fe^3+^, Cr^3+^, and Zr^4+^ were dissolved in 9 mL deionized water to form a concentration of 10^−2^ M solution, which were added dropwise into the previously prepared amPEN aqueous solution and stirred continuously for 10 min, respectively. Finally, amPEN based metallopolymeric superstructures (MPS) were obtained after washing with deionized water three times by centrifuging at 10,000 rpm for 3 min to remove the unreacted polymer, DMF, and ions solution, and dried in an oven at 80 °C for 24 h. MPS was denoted as *x^n+^*-PEN, where *x* and *n* represent different metal ions and corresponding valence, respectively.

### 2.4. Adsorption of Methylene Blue (MB) by MPS

Adsorption of MB dyes from aqueous solution was measured in a batch process by varying initial concentration, contact time, and pH values. In adsorption kinetics model (0–150 min) and pH (2–12) experiments, 20 mL MB dyes aqueous solutions (initial concentration 400 mg/L) containing 15 mg of MPS were stirred at room temperature. The initial pH of MB solution was adjusted to a value of 2–12 by dropwise adding 0.1 mol/L HCl or 0.1 mol/L NaOH solutions. In adsorption isotherms (initial concentration 100–1000 mg/L) experiments, 10 mL of MB dyes aqueous solutions containing 8 mg of MPS were stirred at room temperature. A series of 100–900 mg/L MB aqueous solutions are diluted from MB solution with an initial concentration of 1000 mg/L. The concentration of methylene blue in the supernatant solution before and after adsorption was determined by centrifugation and using a double beam UV spectrophotometer at 664 nm. The amount of dye adsorbed at equilibrium was further determined according to the change of concentration before and after adsorption. The dye adsorption capacity *q_t_* of the MPS for each experiment was calculated according to Equation (1):(1)$$q_{t}=\;\frac{\left(c_{0}-c_{t}\right)V}{W}$$ where *c*_0_ (mg/L) and *c**_t_* (mg/L) are the dye concentration at initial and contact time, respectively, *V* (L) is the volume of dye solution and *W* (mg) is the weight of adsorbent powders. The adsorption capacity (*q_t_)* represents the adsorption amount (mg/g) at an adsorption time of *t* (min) and the equilibrium adsorption capacity (*q_e_*) is defined by same equation using equilibrium concentration *c_e_* (mg/L). 

### 2.5. Characterization

The morphology of the MPS was observed by scanning electron microscopy (SEM, s-4800, Tokyo, Japan) equipped with an energy-dispersive X-ray spectroscope (EDS) operated at an acceleration voltage of 5 kV. X-ray diffraction (XRD) patterns of the MPS powders were collected by a Rigaku-Smartlab diffractometer (RINT2400, Rigaku, Tokyo, Japan) with Cu Kα (λ = 0.15406 nm) radiation in steps of 2° (2*θ*) min^−1^ from 10° to 90°. Molecular structures of synthesized amPEN and MPS were characterized with Fourier transform infrared (FTIR) spectroscopy (470 FTIR, Medison, WI, USA). X-ray photoelectron spectroscopy (XPS) of MPS was recorded on a Thermo Fisher Scientific (XPS, ESCALAB 250Xi XPS, San Jose, CA, USA) with a monochromatic X-ray line source of Al K_α_ radiation. The concentration of dyes in the aqueous solution was acquired from UV-Vis spectrophotometer (TU 1901, Persee, Beijing, China). A ZetaPlus zeta potential analyzer (Brookhaven instruments corporation, New York, NY, USA) were measured the zeta potential of MPS dispersed in the water with different pH at room temperature.

### 2.6. Kinetic Adsorption Isotherm Models

Kinetics of adsorption process were researched by analyzing adsorptive uptake of the dye from solution at different contact time. The pseudo-first-order kinetic model (2), pseudo-second-order kinetic model (3) and the particle internal diffusion model (4) can be fitted using the following Equations (2)–(4):(2)$$\ln\left(q_{e}-q_{t}\right)=\ln q_{t}-K_{1}t$$
(3)$$\frac{t}{q_{t}}=\;\frac{1}{k_{1}q_{e}^{2}}\;+\;\frac{t}{q_{e}}$$
(4)$$q_{t}=\;K_{id}t^{0.5}+\;c_{i}$$

Among them, *q_e_* (mg/g) and *q_t_* (mg/g) are the adsorption capacity of MB adsorbed by the adsorbent, where *q_e_* represents equilibrium adsorption and *q_t_* represents an adsorption time of *t* (min). *K*_1_ (min^−1^) is the adsorption rate constant for a pseudo-first-order kinetic model and *K*_2_ (g/(mg⋅min)) is the pseudo-second-order kinetic model adsorption rate constant. *K_id_* (mg/(g min^−0.5^)) is the internal particle diffusion constant, and *C_i_* is a constant describing the influence of the boundary layer.

Equation (5) of the Langmuir isotherm model and Equation (6) of the Freundlich isotherm model were adopted to investigate the surface properties, adsorbate affinity, and the relationship between the adsorbent and the adsorbate.
(5)$$\ln q_{e}=\;\frac{1}{n}\ln c_{e}+\ln K_{F}$$
(6)$$\frac{c_{e}}{q_{e}}=\;\frac{1}{K_{L}q_{m}}+\;\frac{c_{e}}{q_{m}}$$

In the above equations, *q_m_* (mg/g) represents the maximum adsorption capacity of adsorbent, *K_L_* (L/mg) is the Langmuir constant which is related to the affinity of binding sites, *K_F_* (L/g) is a constant related to the adsorption capacity, *n^−^*^1^ is the adsorption strength, and the remaining parameters are the same as kinetic models. Langmuir isotherm equation defines a dimensionless separation factor (*R_L_*) with the following Equation (7):(7)$$R_{L}=\frac{1}{1+bc_{0}}$$

Among them, *R_L_* is used to indicate the nature of the adsorption process. The calculated result *R_L_* donates the type of the isotherm to be either favorable (0 < *R_L_* <1), irreversible (*R_L_* = 0), reversible (*R_L_* = 1) or unfavorable (*R_L_* > 1).

## 3. Results and Discussion

Polyarylene ether nitrile is a linear thermoplastic that contains abundant aromatic moieties in backbone and polar pendent cyano groups, which contribute to their outstanding thermal stability (glass transition temperature higher than 180 °C), mechanical properties (tensile strength higher than 80 MPa) as well as intrinsic aggregation tendency in concentrated solution derived from π-π interaction between macromolecular chains [38,40]. In order to make the most of their high performance, we have previously introduced the amphipathy feature into PEN macromolecular and fabricated a variety of robust sub-micron superparticles with different morphology, size as well as functionality via the classical oil-in-water microemulsion self-assembly. However, the microemulsion self-assembly protocol still shows disadvantages in terms of low production yield and time-consuming fabrication. For this reason, we have synthesized herein a novel amphiphilic PEN block copolymer containing hydrophobic backbone and hydrophilic sides groups as shown in Figure 1 below. Compared to our previously synthesized PEN, the novel block copolymer bears higher density of hydrophilic carboxyl and sulfonic groups in the side chains, which can contribute to more effective coordination with metal ions, leading to the scalable fabrication of MPS via a facile solvent exchange self-assembly protocol.

Then, a series of metal ions coordinated amPEN were prepared as demonstrated in Figure 2. It can be noticed that the trivalent and tetravalent metal ions rapidly coordinate with amPEN to form visible MPS suspension, while the solution added with divalent metal ions is still clear after 10 min of stirring (see Figure 2a). This dramatic difference should be attributed to the stronger coordination between high valence metal ions and the carboxyl/sulfonic group of amPEN [41]. Moreover, it can be found that Pb^2+^ coordinated amPEN solution also becomes turbid while the solution added with other divalent metal ions are still transparent after 24 h stirring as shown in Figure 2b, indicating that the synthesized amPEN exhibits preferred coordination with Pb^2+^ than the other divalent metal ions [42].

In the above mentioned experiments, we found that the color of amPEN solution in DMF gradually changed to purple during the addition of Fe^3+^ solution, which is different from other samples whose color variation is either derived from added metal ions (Co^2+^, Cr^3+^, Zr^4+^) or possible oxidation of Fe^2+^. It is reported that the color of colloids solution is highly dependent on their morphology [43], thus we assumed that somewhat well-organized microstructures would be formulated following the coordination between Fe^3+^ and synthesized amPEN. Therefore, we have verified the morphology evolution of Fe^3+^ crosslinked amPEN at different experimental conditions. Firstly, it can be discovered from Figure 3 that the concentration of Fe^3+^ has a relatively large influence on the morphology of the complex. It is clear that more regular nanocubes are generated when the amPEN solution is crosslinked by increasing Fe^3+^ content from 10^−3^ M to 5 × 10^−3^ M, while the irregular aggregates are obtained from sample using the highest Fe^3+^ concentration of 10^−2^ M. Since the pH value of Fe^3+^ solution is changed as the variation of its concentration, which will alternate the coordination between metal ions and pendent carboxyl/sulfonate groups of amPEN, thus pH value could be another possible reason for the observed morphology evolution.

Next, we studied the effect of amPEN polymer concentration on the morphology of formulated MPS as demonstrated in Figure 4. It is clear that the more regular and uniform superstructures are created when the concentration of amPEN is less than 50 mg/mL, and the overall size of obtained MPS is increased with the increase of amPEN concentration. It could be explained that the molecular chain can move freely in a dilute solution, while the entanglement of the molecular chain is gradually intensified due to the strong interaction between the molecular chains as the solution concentration increases. However, the macromolecular chain is highly entangled in the concentrated solution (e.g., 100 mg/mL), restricting the organization of amPEN macromolecules, which in turns results in the formation of irregular aggregates as shown in Figure 4d.

Besides the concentration of metal ions and amPEN, we found that the temperature is another important factor to modulate the morphology of obtained MPS. As can be seen in Figure 5, the mixture of various anisotropic nanostructures including nanoplates, nanoprism, and nanorods is obtained when the samples are fabricated at 5 °C (see Figure 5a), while the well-defined nanocubes and nanorods are detected from the sample prepared at 30 °C (see Figure 5b) and 80 °C (see Figure 5c), respectively. However, further increasing the reaction temperature to 120 °C results in the fabrication of irregular nanostructures with curly bundle-like morphology.

Based on the above results, the Fe^3+^ coordinated amPEN basically exhibit anisotropic cube- and rod-like morphology regardless of reactants concentration and reaction temperature, it would be interesting to fabricate MPS with more abundant morphological features. Given that the initially transparent amPEN solution can also get turbid after complexing with other metal ions such as Pb^2+^, Cr^3+^, and Zr^4+^ (see Figure 2), we further characterized the morphology and structures of these metal ions coordinated amPEN under optimized experimental conditions with amPEN concentration of 50 mg/mL and reaction temperature of 30 °C. Basically, the isotropic nanospheres are obtained from divalent Pb^2+^ coordinated amPEN (see Figure 6a), the typical nanocubes are obtained for trivalent Fe^3+^ and Cr^3+^ coordinated amPEN (see Figure 6b,c), while more complex star-like superstructures are generated when amPEN is crosslinked with tetravalent Zr^4+^ as shown in Figure 6d. Furthermore, energy-dispersive X-ray spectroscopy (EDS) techniques have been employed to confirm the presence of metal ions in the generated MPS. As shown in Figure 6a_0_-d_0_, typical peaks corresponding to lead, iron, chrome, and zirconium elements appear in the EDS spectrum of Pb^2+^-PEN, Fe^3+^-PEN, Cr^3+^-PEN and Zr^4+^-PEN, respectively, which confirms that Pb^2+^, Fe^3+^, Cr^3+^, and Zr^4+^ are successfully introduced into corresponding MPS.

Then, we recorded the XRD patterns of as-synthesized amPEN powder, Pb^2+^-PEN, Fe^3+^-PEN, Cr^3+^-PEN and Zr^4+^-PEN metallopolymeric superstructures (MPS) as shown in Figure 7. It is clear that the initially amorphous amPEN (see Figure 7a) can be transformed into highly crystallized MPS as illustrated in Figure 7b. Generally, the higher valency ions of Fe^3+^, Cr^3+^, and Zr^4+^ coordinated amPEN demonstrate more or less the same diffraction peaks with slightly different intensity, while Pb^2+^ coordinated amPEN sample exhibits more diffraction peaks that are not located in the same position as those of other samples. The dramatic difference between Pb^2+^ coordinated amPEN and other samples is also consistent with the morphology characterization in Figure 6, where the former one is isotropic nanospheres and the others basically are anisotropic superstructures. Although the specific reason for this difference is still unclear, we assume that the Pb^2+^ coordinated amPEN could be generated from the self-assembly of colloidal building blocks with different crystallization.

In order to further explore the mechanism of metal ions coordination with amPEN, XPS spectroscopy was used to analyze the samples. As shown in Figure 8a, the survey spectra show that all the expected elements, such as C, O, N, and S, are from amPEN and metal ions are from metallic compounds, which confirmed that metal ions have been introduced into MPS successfully. Figure 8b shows the high resolution XPS of C1s and there are four peaks in the spectrum. Taking Fe^3+^-PEN as an example, the binding energy at 284.7 eV corresponds to the C–C or C=C group and the binding energy at 285.7 eV is ascribed to the C≡N group, and the binding energy of 286.6 eV and 287.6 eV is attributed to the C–O group and the C=O group, respectively [44]. Figure 8c shows the high resolution XPS of O1s again using Fe^3+^-PEN as an example, where the typical peak corresponding to the S=O or C=O, C–O–C and O–C=O groups is detected at 531.9 eV, 531.1 eV, and 532.7 eV, respectively [45]. Finally, the high-resolution spectra of Pb 4f, Fe 2p, Cr 2p, and Zr 3d peaks are displayed in Figure 8d–g, respectively. The Pb 4f spectrum peaked at 138.8 and 143.7 eV, which could be assigned to Pb 4f_7/2_ and Pb 4f_5/2_, respectively, and previous studies have reported that the typical binding energy of O-Pb in complex located at ~143 eV [46,47]. Similarly, the Fe 2p_3/2_ peaked at 710.1 eV, the Cr 2p_3/2_ peaked at 576.9 eV and the Zr 3d_3/2_ peaked at 182.3 eV is ascribed to O–M (M = Fe, Cr or Zr) in complex [48,49,50]. Therefore, these XPS results unambiguously confirm that the metal ions are coordinated with pendent carboxyl or sulfonic groups of synthesized amPEN block copolymer.

Moreover, the FTIR spectra of obtained MPS are shown in Figure 9, where the peak at 1718 cm^−1^ corresponds to the C=O adsorption of the free carboxyl group and the adsorption peak at 1193 cm^−1^ results from the S=O asymmetric stretching vibration of the sulfonic acid group. The characteristic peak of 1648~1658 cm^−1^ from Pb^2+^, Fe^3+^, Cr^3+^, and Zr^4+^ coordinated amPEN is obviously shifted as compared to that of pristine amPEN, mainly due to the alternation of antisymmetric stretching vibration peak of carboxyl group after coordination. Alternatively, the adsorption peak at 1384 cm^−1^ is detected, which should be a result from the typical symmetric stretching vibration (COO^−^). These results unambiguously confirm that amPEN is successfully coordinated with Pb^2+^, Fe^3+^, Cr^3+^, and Zr^4+^.

Based on the above experiments, we have confirmed that the amorphous amphiphilic PEN block copolymer can be transformed into crystallized metallopolymeric superstructures (MPS) with different morphology. It should be noted that the fabrication of MPS is facile and easy to be scaled up, thus could be employed as cost-effective adsorbents for dyes removal. Therefore, we chose the widely used methylene blue (MB) as the model dye to evaluate the adsorption performance of Pb^2+^-PEN, Fe^3+^-PEN, Cr^3+^-PEN, and Zr^4+^-PEN MPS. It is clear from Figure 10a that the adsorption capacity increases with the extension of adsorption time and reached equilibrium about at 60 min for all the MPS samples in the kinetics experiments. More specifically, a higher adsorption rate is recorded at beginning, which should be attributed to the fact that the adsorption sites of adsorbents are initially available. Afterward, the adsorption rate is lower due to the decrease of vacant sites of adsorbents as well as dye concentration. The decreased adsorption rate, especially when the experiment is approaching to the end, indicates that a single layer of MB may form on the surface of the adsorbents, shielding the effective sites required for further adsorption of dye molecules after reaching equilibrium. In addition, we can see that the adsorption effect of the four adsorbents on MB is different, and the adsorption capacity of Pb^2+^-PEN and Cr^3+^-PEN is greater than that of Fe^3+^-PEN and Zr^4+^-PEN, which is due to the obvious difference in specific surface area. The larger the specific surface area, the faster the adsorption rate and the larger the adsorption capacity, which is consistent with the SEM characterization in Figure 6.

The pseudo-first-order and pseudo-second-order kinetic models were used to analyze the kinetics of the adsorption process. The theoretical adsorption capacity of four MPS adsorbents calculated from the pseudo-first-order model and pseudo-second-order model were summarized in Table 1. As shown in Figure 10b,c, the *R*^2^ values obtained from the pseudo-second-order model exhibited better linearity than those obtained from the pseudo-first-order model. At the same time, the theoretical equilibrium adsorption amount *q_e_* calculated by the pseudo second-order kinetic model is closer to the experimentally obtained actual equilibrium adsorption amount *q_e,exp_*. In conclusion, the kinetic model shows that the adsorption rate of MB on Pb^2+^-PEN, Cr^3+^-PEN, Fe^3+^-PEN, and Zr^4+^-PEN can be easily affected by chemisorption.

It can be seen from Figure 10d and Table 2 that the intraparticle diffusion fitting curves of MB dyes adsorption by Pb^2+^-PEN, Cr^3+^-PEN, Fe^3+^-PEN, and Zr^4+^-PEN adsorbents are multi-segmented, which implies that intraparticle diffusion is not the only step to determine the adsorption reaction rate. In the first stage, the rapid adsorption phase within 10 min is the process of out-diffusion control. In the second stage, the adsorption phase of 10–60 min gradually increases, which is the diffusion adsorption process of the dye in the pores of the adsorbent. After 60 min, the adsorption rate is very low due to reaching adsorption equilibrium. Therefore, the adsorption rate of the three stages is *K*_1_ > *K*_2_ > *K*_3_, and as the adsorption process progresses, adsorption position of the dye on the adsorbent increases, the boundary thickness gradually increases, so *c*_1_ < *c*_2_ < *c*_3_.

The effect of initial concentration of MB solution on the adsorption performance of Pb^2+^-PEN, Cr^3+^-PEN, Fe^3+^-PEN, and Zr^4+^-PEN MPS are displayed in Figure 11. Generally, all the obtained adsorption capacity of MPS exhibit similar trends with the increasing of MB concentration. It is observed that, when the initial concentration of MB is less than 500 mg/L, the adsorption capacity increases linearly with the increase of the initial concentration, which can be explained by there not being enough dyes molecules to fill all available vacant adsorption sites. However, when the initial concentration of MB is above 500 mg/L, the adsorption capacity gradually tends to balance out with the increase of MB concentration; this is because the adsorption sites on the MPS surface are not enough and the MB needs overcome the mass transfer resistance.

The amount of MB adsorbed per unit of mass of samples and the equilibrium concentration in the aqueous solution at room temperature is presented in Figure 12a. The maximum adsorption capacities of Pb^2+^-PEN, Cr^3+^-PEN, Fe^3+^-PEN, and Zr^4+^-PEN for MB are determined to be 936.13 mg/g, 871.67 mg/g, 889.08 mg/g, and 769.44 mg/g, respectively, which are basically larger than those of recently reported polymeric nanostructures adsorbents by other groups as demonstrated in Table 3. To investigate the interaction of adsorbate molecules and adsorbent surface, two well-known models, Langmuir and Freundlich isotherms, are explored to explain dye adsorbent interaction in this study. From Figure 12b,c and Table 4, the linear correlation coefficient *R*^2^ of the Langmuir isotherm adsorption model for MB adsorption is greater than 0.999, and the linear correlation coefficient R^2^ of the Freundlich isotherm adsorption model is less than 0.922. This behavior is consistent with the Langmuir isotherm adsorption model and is a single-layer adsorption. In order to further investigate whether the adsorption of MB by adsorbent is easy to occur, the Langmuir adsorption equilibrium constant *K_L_* at room temperature and the different initial concentration *c**_0_* of MB solution are calculated according to Equation (7) to obtain the separation factor *R_L_*, which is related to *c**_0_*. It can be seen from Figure 12d that the *R_L_* value of the adsorbent to MB is less than 1, indicating that the adsorbent is easy to adsorb the dye MB.

Finally, the pH of the solution is known to affect adsorption performance by changing the surface charge of the adsorbent. Figure 13a shows that when the pH is increased from 2 to 12, the adsorbed amount of MB by MPS gradually increased; Pb^2+^-PEN, Cr^3+^-PEN, Fe^3+^-PEN, and Zr^4+^-PEN show a similar trend. The free carboxyl/sulfonic groups of obtained MPS are deprotonated in basic solution, which will facilitate the adsorption of positively charged MB via electrostatic interaction. On the contrary, high H^+^ concentrations in acid solution will strongly compete with MB cations for available adsorption sites of MPS. The characterization on the zeta potential of MPS adsorbents can further confirm the above conclusions, as shown in Figure 13b. It can be seen that the zeta potential of MPS adsorbents are negative even under the condition of strong acid, which can offset effects of the protonation of the strong acid and obtain a great adsorption capacity. Simultaneously, it can be seen that the zeta potentials of MPS under alkaline conditions is significantly lower than acidic conditions. MB as a cationic dye, when the zeta potential is lower, the adsorption is better. It proves that the adsorption ability under alkaline conditions is better than acidic conditions. Therefore, electrostatic interaction is the primary driving force for the removal of MB by the obtained MPS adsorbents.

## 4. Conclusions

In this work, a novel amphiphilic block copolymer composed of hydrophobic backbone and hydrophilic pendent groups has been synthesized and transformed into crystallized metallopolymeric superstructures via a facile metal ions-mediated solvent exchange self-assembly. It was found that the obtained MPS exhibit different morphology including isotropic nanospheres as well as anisotropic nanocubes, nanorods, and nanostars depending on the coordinating metal ions. Meanwhile, the Pb^2+^, Fe^3+^, Cr^3+^, and Zr^4+^ coordinated MPS demonstrates competitive adsorption capacity towards cationic dye of MB with maximum adsorption capacities up to 936.13, 871.67, 889.08, and 769.44 mg/g, respectively. Kinetics experiments reveal that the equilibrium adsorption is practically achieved in 60 min, intraparticle diffusion is not the only step to determine the adsorption reaction rate, and adsorption behavior can be described by a monolayer Langmuir type isotherm. In addition, all the obtained MPS adsorbents demonstrate good stability and MB removal capacity in a wide pH range from 2 to 12. Given the facile fabrication, versatile morphology, and promising MB adsorption capacity of these metallopolymeric superstructures, we assume that the present work open new ways for the environmental application of high-performance polyarylene ethers thermoplastics.

## Figures and Tables

**Figure 1 nanomaterials-09-01001-f001:**
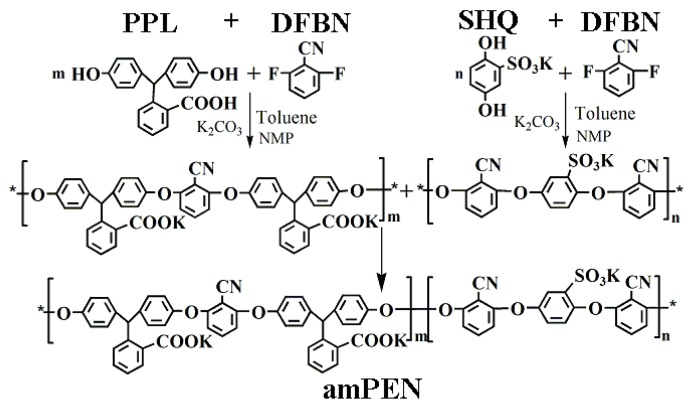
The synthesis route for amphiphilic PEN block copolymer.

**Figure 2 nanomaterials-09-01001-f002:**
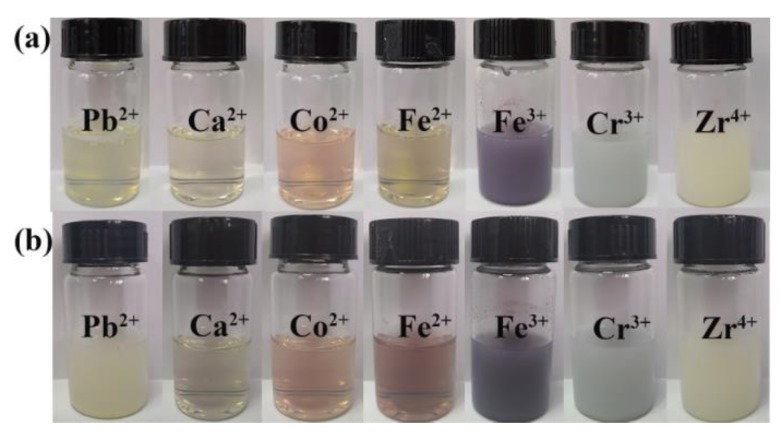
Sample vials containing DMF (1 mL) solution of amPEN (50 mg) added with different metal ions aqueous solution (10^−2^ M) after stirring for 10 min (**a**) and 24 h (**b**).

**Figure 3 nanomaterials-09-01001-f003:**
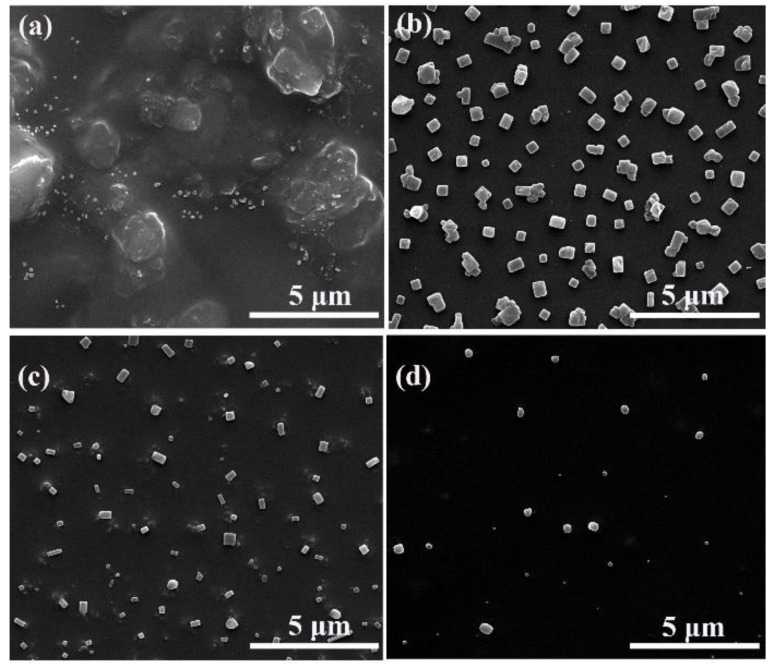
SEM images of metallopolymeric superstructures (MPS) formulated by adding (**a**) 10^−2^ M, (**b**) 5 × 10^−3^ M, (**c**) 2.5 × 10^−3^ M and (**d**) 10^−3^ M aqueous aliquots of Fe^3+^ into amPEN solution in DMF (50 mg/mL).

**Figure 4 nanomaterials-09-01001-f004:**
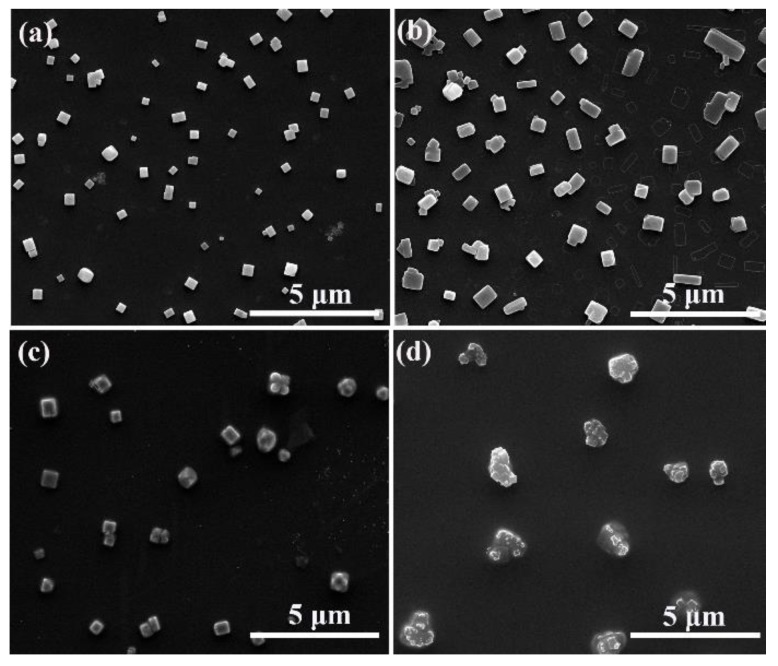
SEM images of MPS obtained by adding Fe^3+^ aqueous solution (5 × 10^−3^ M) into DMF solution dissolved different contents of synthesized amPEN copolymer of (**a**) 25 mg/mL, (**b**) 50 mg/mL, (**c**) 75 mg/mL and (**d**) 100 mg/mL, respectively.

**Figure 5 nanomaterials-09-01001-f005:**
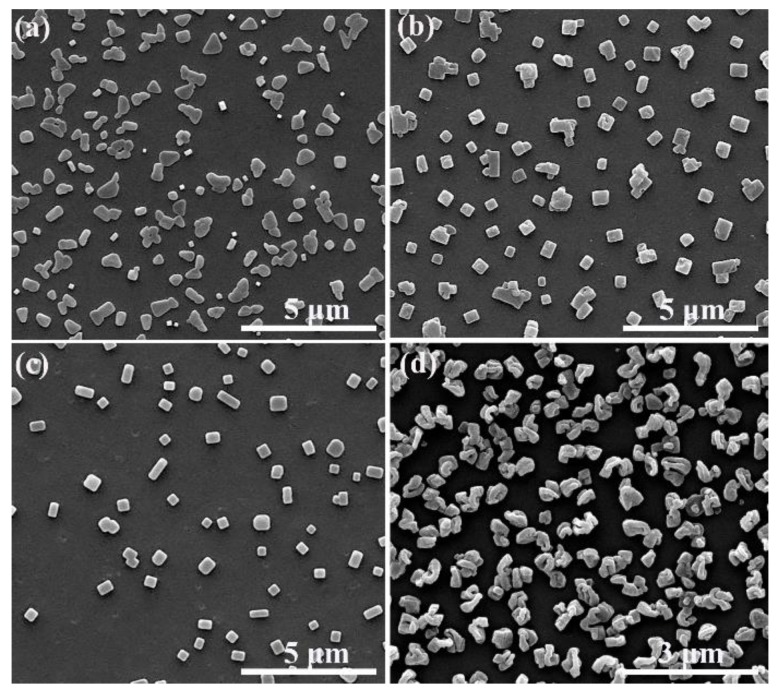
SEM images of MPS obtained via the coordination of Fe^3+^ (5 × 10^−3^ M) and amPEN (50 mg/mL) at different temperature of (**a**) 5 °C, (**b**) 30 °C, (**c**) 80 °C, and (**d**) 120 °C, respectively.

**Figure 6 nanomaterials-09-01001-f006:**
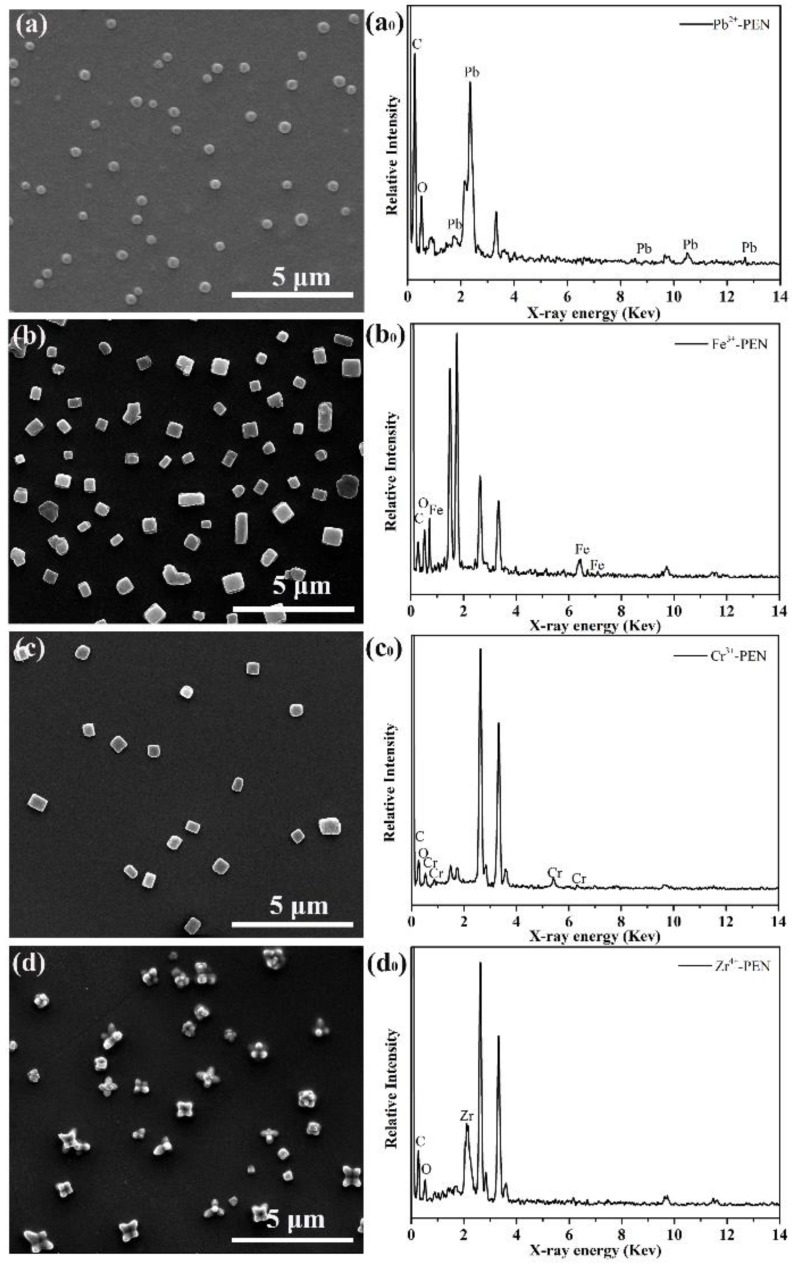
SEM images (left column) and corresponding EDS spectrum (right column) of MPS formulated by crosslinking amPEN with (**a**,**a_0_**) Pb^2+^ (**b**,**b_0_**) Fe^3+^, (**c**,**c_0_**) Cr^3+^ and (**d**,**d_0_**) Zr^4+^, respectively.

**Figure 7 nanomaterials-09-01001-f007:**
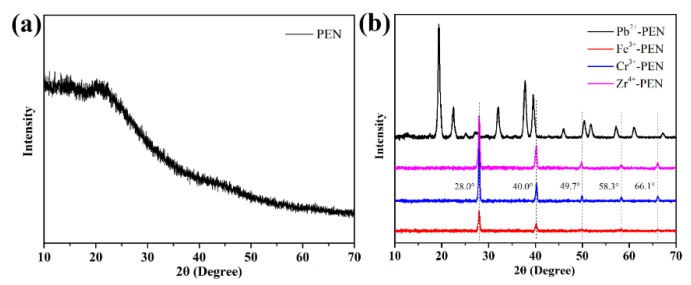
XRD patterns of as-synthesized amPEN powder (**a**) and formulated metal ions coordination polymeric superstructures (**b**).

**Figure 8 nanomaterials-09-01001-f008:**
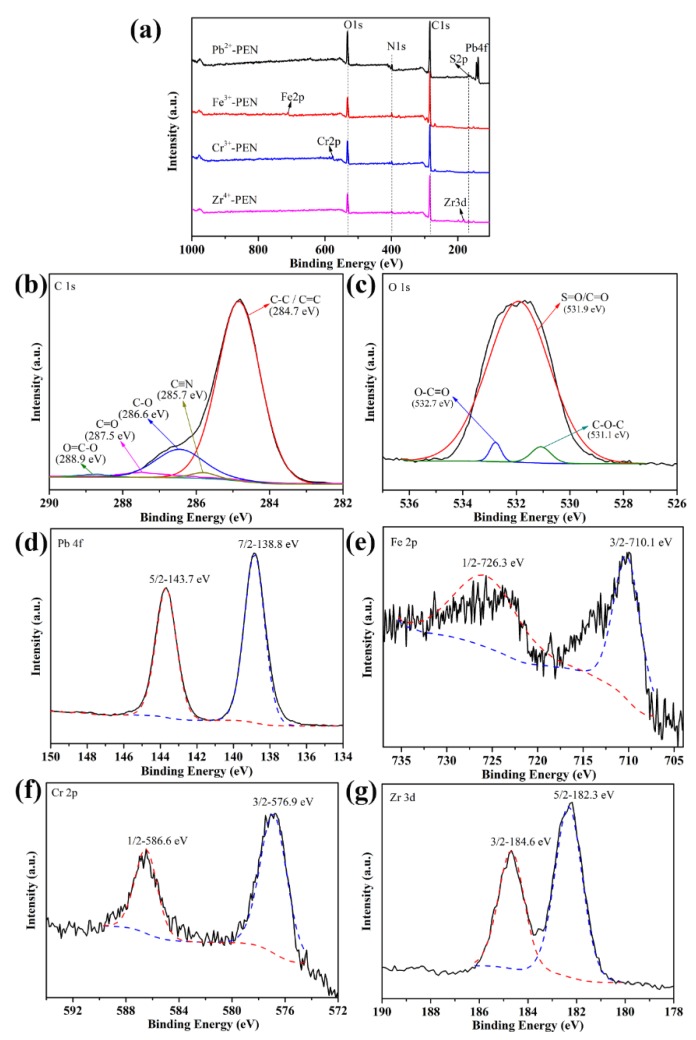
High-resolution X-ray photoelectron spectroscopy (XPS) spectra of MPS (**a**), elemental spectrum corresponding to C 1s (**b**), O 1s (**c**) of Fe^3+^-PEN MPS as well as Pb 4f (**d**), Fe 2p (**e**), Cr 2p (**f**), and Zr 3d (**g**) of MPS.

**Figure 9 nanomaterials-09-01001-f009:**
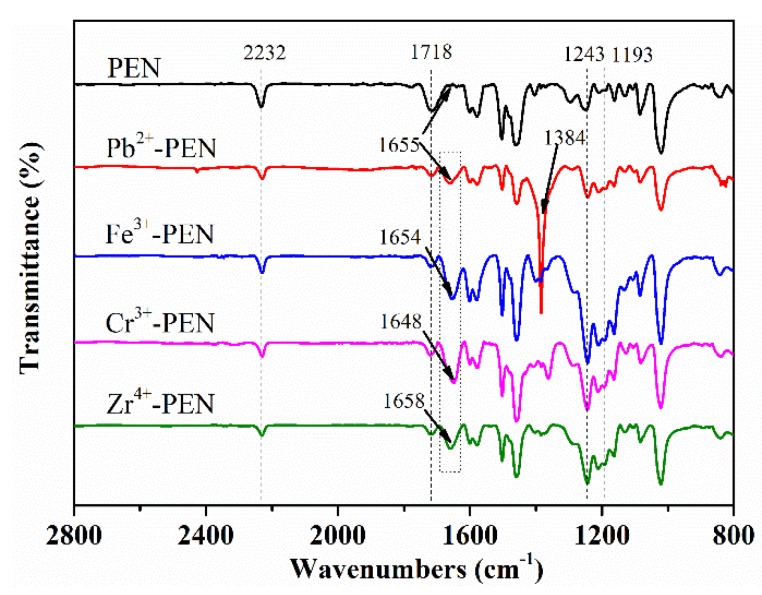
The FTIR spectra of amPEN superstructures coordinated with different metal ions.

**Figure 10 nanomaterials-09-01001-f010:**
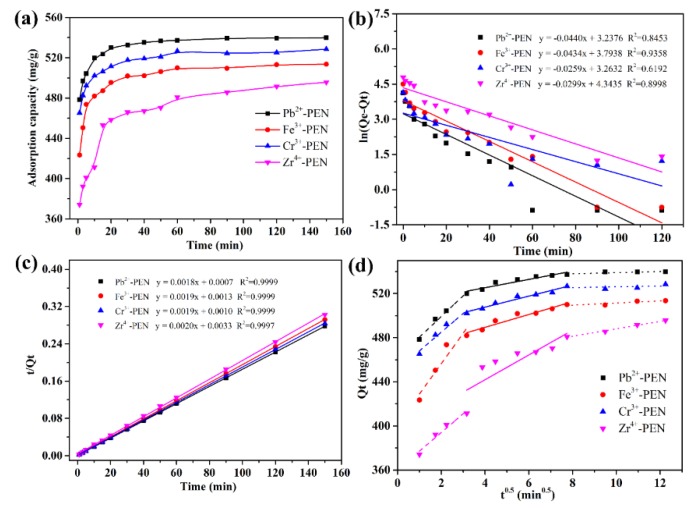
The adsorption performance of obtained MPS towards methylene blue (MB) dyes: variation of adsorption capacity with adsorption time (**a**), pseudo-first-order kinetics (**b**), pseudo-second-order kinetics (**c**), and the internal diffusion equation of particles for adsorption of MB (**d**).

**Figure 11 nanomaterials-09-01001-f011:**
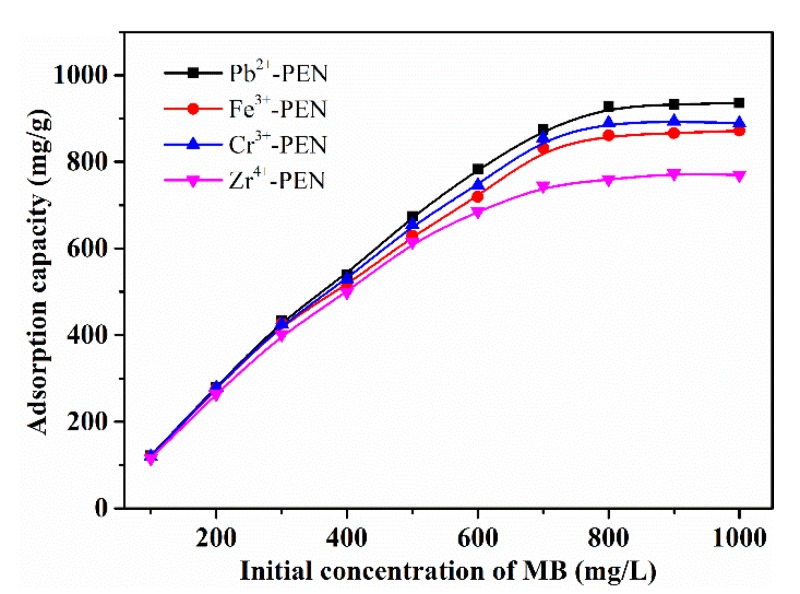
Effect of initial MB concentration on the adsorption capacity of MPS adsorbent.

**Figure 12 nanomaterials-09-01001-f012:**
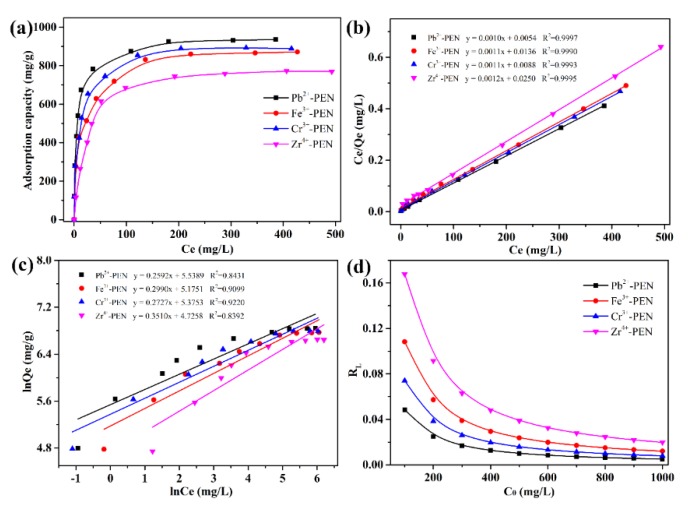
(**a**) Adsorption isotherm of MB solution; (**b**) Langmuir isotherm plot for MB adsorption; (**c**) Freundlich isotherm plot for MB adsorption; (**d**) The relation curves of *R_L_* versus *c**_0_*.

**Figure 13 nanomaterials-09-01001-f013:**
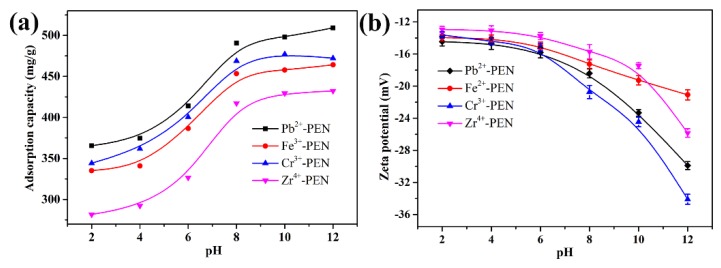
(**a**)The adsorbed amount of MB by obtained MPS under different pH values; (**b**) Zeta potential of the MPS at different pH values. Error bars represent the standard deviations of three individual experiments.

**Table 1 nanomaterials-09-01001-t001:** Parameters of pseudo-first-order kinetic model and pseudo-second-order kinetic model for the adsorption of MB by different MPS adsorbents.

Entry	*Q_e,exp_*	Pseudo-First-Order Model			Pseudo-Second-Order Model		
		*K* _1_	*q_e,cal_*	*R* ^2^	*K_2_*	*q_e,cal_*	*R* ^2^
	(mg/g)	(min^−1^)	(mg/g)		(g/(mg·min))	(mg/g)	
Pb^2+^-PEN	539.89	0.0440	25.473	0.8453	0.0046	555.56	0.9999
Fe^3+^-PEN	513.57	0.0434	44.425	0.9358	0.0028	526.32	0.9999
Cr^3+^-PEN	528.51	0.0259	26.133	0.6192	0.0036	526.32	0.9999
Zr^4+^-PEN	495.67	0.0299	76.976	0.8998	0.0012	500.00	0.9997

**Table 2 nanomaterials-09-01001-t002:** Fitting parameters of the internal diffusion equation of MPS adsorbents.

Entry	0~10 min			10~60 min			60~150 min		
	*K* _1_	*c_i_*	*R*	*K* _2_	*c_i_*	*R* ^2^	*K_id_*	*c_i_*	*R* ^2^
Pb^2+^-PEN	18.76	461.69	0.9974	3.77	510.24	0.8880	0.54	533.61	0.9918
Fe^3+^-PEN	27.56	401.35	0.8625	5.82	466.12	0.9262	0.95	501.88	0.9940
Cr^3+^-PEN	16.98	451.06	0.9417	5.01	487.66	0.9579	0.35	522.67	0.9915
Zr^4+^-PEN	16.96	360.15	0.9372	11.23	396.96	0.6665	3.36	454.40	0.9904

**Table 3 nanomaterials-09-01001-t003:** Adsorption performance of recently reported polymeric nano-adsorbents.

Kinds of Adsorbents	Adsorption Capacity	Temperatures	Reference
Poly(cyclotriphosphazene-co-phloroglucinol) (PCCP) microspheres	50.7 mg/g	298 K	[51]
Porous poly-melamine-formaldehyde (PMF)	82.5 mg/g	298 K	[52]
Polydopamine (PDA) microspheres	90.7 mg/g	298 K	[53]
Porous Poly(imide-ether)s (PIEs)	166.8 mg/g	303 K	[54]
Bakelite-type anionic microporous organic polymers (MOPs)	712.2 mg/g	298 K	[55]
Magnetic adsorbent (Na-(CS/PAA)n/MPC)	305.8 mg/g	298 K	[56]
amphoteric β-cyclodextrin-based adsorbent	335.5 mg/g	298 K	[57]
Pb^2+^-PEN	936.13 mg/g	298 K	This work
Fe^3+^-PEN	871.67 mg/g	298 K	This work
Cr^3+^-PEN	889.08 mg/g	298 K	This work
Zr^4+^-PEN	769.44 mg/g	298 K	This work

**Table 4 nanomaterials-09-01001-t004:** Parameters for Langmuir and Freundlich adsorption isotherm models.

Entry	*q_m,exp_*	Langmuir Model			Freundlich Model		
	(mg/g)	*K_L_* (L/mg)	*q_m_* (mg/g)	*R* ^2^	*K_F_* (L/g)	*n*	*R* ^2^
Pb^2+^-PEN	936.13	0.1963	943.40	0.9997	254.40	3.8580	0.8431
Fe^3+^-PEN	871.67	0.0824	892.86	0.9990	176.81	3.3444	0.9099
Cr^3+^-PEN	889.08	0.1250	909.10	0.9993	216.00	3.6670	0.9220
Zr^4+^-PEN	769.44	0.0496	806.45	0.9995	112.82	5.6252	0.8392

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
