# Peer review of "Scalable Fabrication of Metallopolymeric Superstructures for Highly Efficient Removal of Methylene Blue"

_nanomaterials, 2019, doi:10.3390/nano9071001_

Reviewer 1 Report

The manuscript focuses on the elimination of methylene blue (MB) from aqueous environment using a new kind of amphiphilic metallopolymeric superstructures (MPS) as adsorbent. The topic, i.e. elimination of dyes from water sources is an urgent task nowadays. The authors describe the synthesis steps of a new type amphiphilic polymers and the preparation of metallopolymeric superstructures. Throughout the paper they quantitatively characterize the synthesized materials and study the kinetics of the adsorption of MB on the MPS. The effect of concentration, temperature on the MPS morphology and the influence of pH on the MB adsorption on MPS are also discussed in the manuscript. Although the experimental work is extensively performed, the text is well written and the graphs support the understanding of the scientific content of the paper, some questions arose during reading:

Comments, remarks:

Adsorption and absorption are different phenomena, they cannot be used as synonyms of each other. Chemical species can adsorb on a surface, while e.g. light can be absorbed by materials resulted in various colours. This manuscript discuss the adsorption of MB on MPS, so please replace all „absorption” or „absorbent” to „adsorption” and „adsorbent”.

Line 74: in vitro is usually written in Italics

Line 88: „ferrous chloride” instead of „ferrouschloride”

Line 87-89: metal salts used in the experiments are with or without crystal water?

Line 98: At what pH did you mix the components? Did you adjusted the pH? What can be the effect of the difference in pH of the various metal ion solutions?

Line 99: What does it mean amphiphilic? If it is really amphiphilic, it shoud be soluble in water too. Why did you dissolve it in DMF?

Line 102: Did all the metal ions remain immobilized in the MPS? Did you measure the possible relaease/leakage of metal ions?

Line 108: Is it the absorbance of only MB? May the possibly released influence the measured absorbance? Because they have characteristic color.

Line 113: concentration is usually not abbreviated in capital

Line 115: Why did you define two adsorption capacity values? What is the difference between them? Later on it becomes more clear, but maybe you should mention it here/in this paragraph as well.

Line 156-157: The sentence is not clear – please re-create it

Line 194: Maybe this effect can be attributed not only to the concentration - with the concentration, the pH is also changing.

Line 291: FTIR instead of FIIR

Line 356: At what pH were these adsoprtion isotherms determined?

Line 369: „MB adsorption” instead of „MB adsorbent adsorption”

Line 380, 389: „adsorbed amount” instead of „adsorption amount”

Line 381: „are deprotonated” instead of „is de-protonated”

Line 386: „driving force” instead of „driven force”

Line 389 (and Fig 13.): is it in the function of time or of the pH?

Line 398: „adsorption equilibrium is” instead of „equilibrium adsorption are”

Line 401: „good stability” – How dou you mean? Colloid stability? Chemical stability? Shelf-life?

Line 404: „open new ways”

Line 407: „contributed to…”

Reviewer 2 Report

The manuscript entitled “Scalable fabrication of metallopolymeric superstructures for highly efficient removal of methylene blue” submitted by  Zhou, Jia, Liu and coworkers. Presents the synthesis and characterization and application for Methylene blue absorption of  several metallopolymeric superstructures (MPS). Specifically, fist is described the synthesis of amphiphilic block copolymers (BCP), and its subsequently transformed into MPS. Procedures are well described and the materials are sufficiently characterized. Therefore I find this manuscript suitable for publication after addressing the next few issues:

The effect of metal in methylene blue is not well described. First, some interaction model could be proposed. Into the same line, methylene blue uptake of non-metalated polymer should also be presented.

Since good values of methylene blue adsorption are described, porosity measurements and BET surface area measurements would greatly appreciated.

Finally, the specificity issue is not addressed. Adsorption is good for other pigments or just for methylene blue? There is some selectivity phenomena?

After addressing these issues I fell that the manuscript suitable for publication.

Reviewer 3 Report

Synthesis of amphiphilic block copolymer is one of the most important part in this work. The reference used to indicate their previous work is not right. The authors must take special care while putting the references. The physical properties of the synthesized MPS need to mention.

The authors mentioned the MPS as ‘absorbent’, but the removal process as ‘adsorption’. Please note that the terms ‘absorption’ and ‘adsorption’      indicate different mechanism. The authors need to correct and explain it in the manuscript.

In section 2.3: The authors mixed the polymer solution in DMF and aqueous metal cation solution. What is the solubility of the polymer in water? The      author should provide a picture with polymer solution and water mixture.

In section 2.3, line 98: “…different amounts of synthesized amPEN were dissolved…” It is not clear why did the authors use different concentration…Please indicate the metal ion and polymer concentration.

In line 101: The authors mentioned “triple centrifugation washing…”. This      part is not clear. Did the authors re-dissolved the centrifuged polymer? Please also indicate the drying method.

In section 2.4: Please write the dye removal experimental detail.

For line 159-163: Please provide the reference.

In Figure 1: Please put the compound name where applicable.

In line 179-180: “This dramatic difference should be attributed to the      stronger coordination between high valence metal ions and the      carboxyl/sulfonic group of amPEN” and in line 181-183: “…solution also      becomes turbid… exhibits preferred coordination…” Please      provide the reference. Also in line 190, for “It is reported that…”

Please try to use passive voice where appropriate.

How did the authors prepare the samples for SEM? Is it from the solution or      from dry state?

In Figure 3: Please indicate the concentration of the amPEN solution.

In Figure 4: Please use the abbreviation ‘MPS’. Please indicate the Fe ion      concentration.

Please provide high resolution SEM image of the MPS to demonstrate its surface      nature.

The authors need to study the stability of the MPS structure after the      experiment. The aqueous solution may change the structure of the MPS. The      SEM image could be useful to show any physical changes.

In Table 3: Please also add the adsorption capacity of some low cost      adsorbents (mentioned in line 56 and 57). Please confirm, whether these are      absorbents or adsorbents?

In Figure 13: Please correct the x axis. Please also indicate the zero point charge      of the corresponding MPS that may help to understand the pH effect better.

Author Response

Round  2

Reviewer 3 Report

The authors responded "The synthesized amPEN is hardly dissolved in pure water as shown in Figure R2a." They provide the polymer solution picture in 9:1 DMF:water solution. However, in revised manuscript, they prepared the solution with 1:9 DMF:water mixture. The solubility of the polymer in 1:9 DMF-water is highly questionable.

In line 103: The authors mentioned “triple centrifugation washing…”. What does this 'triple centrifugation washing' mean?

Author Response

Round  3

Reviewer 3 Report

The authors revised the work as per Referee's comments and suggestions. The comments are explained quite satisfactorily. The article seems suitable for the publication.

Nanomaterials EISSN 2079-4991 Published by MDPI AG, Basel, Switzerland RSS E-Mail Table of Contents Alert
Back to Top